# GRAFIMO: Variant and haplotype aware motif scanning on pangenome graphs

**Manuel Tognon**[1], **Vincenzo Bonnici**[1], **Erik Garrison**[2], **Rosalba Giugno**[1]*,
**Luca Pinello**[3,4,5]*

**1** Computer Science Department, University of Verona, Verona, Italy, **2** University of Tennessee Health Science Center, Memphis, Tennessee, United States of America, **3** Molecular Pathology Unit, Center for Computational and Integrative Biology and Center for Cancer Research, Massachusetts General Hospital Charlestown, Massachusetts, United States of America, **4** Department of Pathology, Harvard Medical School, Boston, Massachusetts, United States of America, **5** Broad Institute of MIT and Harvard, Cambridge, Massachusetts, United States of America

* rosalba.giugno@univr.it (RG); lpinello@mgh.harvard.edu (LP)

**Data Availability Statement:** All relevant data are within the manuscript and its Supporting Information files.

**Funding:** LP was supported by National Human Genome Research Institute R00HG008399 and

## Abstract

Transcription factors (TFs) are proteins that promote or reduce the expression of genes by binding short genomic DNA sequences known as transcription factor binding sites (TFBS). While several tools have been developed to scan for potential occurrences of TFBS in linear DNA sequences or reference genomes, no tool exists to find them in pangenome variation graphs (VGs). VGs are sequence-labelled graphs that can efficiently encode collections of genomes and their variants in a single, compact data structure. Because VGs can losslessly compress large pangenomes, TFBS scanning in VGs can efficiently capture how genomic variation affects the potential binding landscape of TFs in a population of individuals. Here we present GRAFIMO (GRAph-based Finding of Individual Motif Occurrences), a command-line tool for the scanning of known TF DNA motifs represented as Position Weight Matrices (PWMs) in VGs. GRAFIMO extends the standard PWM scanning procedure by considering variations and alternative haplotypes encoded in a VG. Using GRAFIMO on a VG based on individuals from the 1000 Genomes project we recover several potential binding sites that are enhanced, weakened or missed when scanning only the reference genome, and which could constitute individual-specific binding events. GRAFIMO is available as an open-source tool, under the MIT license, at https://github.com/pinellolab/GRAFIMO and https://github.com/InfOmics/GRAFIMO.

## Author summary

Transcription factors (TFs) are key regulatory proteins and mutations occurring in their binding sites can alter the normal transcriptional landscape of a cell and lead to disease states. Pangenome variation graphs (VGs) efficiently encode genomes from a population of individuals and their genetic variations. GRAFIMO is an open-source tool that extends the traditional PWM scanning procedure to VGs. By scanning for potential TBFS in VGs, GRAFIMO can simultaneously search thousands of genomes while accounting for SNPs,

Genomic Innovator Award R35HG010717. RG was supported from the European Union's Horizon 2020 research and innovation programme under grant agreement 814978 and JPcofuND2 Personalised Medicine for Neurodegenerative Diseases project JPND2019-466-037. The funders had no role in study design, data collection and analysis, decision to publish, or preparation of the manuscript.

**Competing interests:** The authors have declared that no competing interests exist.

indels, and structural variants. GRAFIMO reports motif occurrences, their statistical significance, frequency, and location within the reference or alternative haplotypes in a given VG. GRAFIMO makes it possible to study how genetic variation affects the binding landscape of known TFs within a population of individuals.

This is a *PLOS Computational Biology* Software paper.

## Introduction

Transcription factors (TFs) are fundamental proteins that regulate transcriptional processes. They bind short (7-20bp) genomic DNA sequences called transcription factor binding sites (TFBS) [1]. Often, the binding sites of a given TF show recurring sequence patterns, which are referred to as motifs. Motifs can be represented and summarized using Position Weight Matrices (PWMs) [2], which encode the probability of observing a given nucleotide in a given position of a binding site. In recent years, several tools have been proposed for scanning regulatory DNA regions, such as enhancers or promoters, with the goal of predicting which TF may bind these genomic locations. Importantly, it has been shown that regulatory motifs are under purifying selection [3,4], and mutations occurring in these regions can lead to deleterious consequences on the transcriptional states of a cell [5]. In fact, mutations can weaken, disrupt or create new TFBS and therefore alter expression of nearby genes. Mutations altering TFBS can occur in haplotypes that are conserved within a population or private to even a single individual, and can correspond to different phenotypic behaviour [6,7]. For these reasons, population-level analysis of variability in TFBSs is of crucial importance to understand the effect of common or rare variants to gene regulation. Recently, a new class of methods and data structures based on genome graphs have enabled us to succinctly record and efficiently query thousands of genomes [8]. Genome graphs optimally encode shared and individual haplotypes based on a population of individuals. An efficient and scalable implementation of this approach called variation graphs (VGs) has been recently proposed [9]. Briefly, a VG is a graph where nodes correspond to DNA sequences and edges describe allowed links between successive sequences. Paths through the graph, which may be labelled (such as in the case of a reference genome), correspond to haplotypes belonging to different genomes [10]. Variants like SNPs and indels form bubbles in the graph, where diverging paths through the graph are anchored by a common start and end sequence on the reference [11]. VGs offer new opportunities to extend classic genome analyses originally designed for a single reference sequence to a panel of individuals. Moreover, by encoding individual haplotypes, VGs have been shown to be an effective framework to capture the potential effects of personal genetic variants on functional genomic regions profiled by ChIP-seq of histone marks [12]. During the last decade, several methods have been developed to search TFBS on linear reference genomes, such as FIMO [13] and MOODS [14] or to account for SNPs and short indels such as is-rSNP, TRAP and atSNP [15–17], however these tools do not account for individual haplotypes nor provide summary on the frequency of these events in a population. To solve these challenges, we have developed GRAFIMO, a tool that offers a variation- and haplotype-aware identification of TFBS in VGs. Here, we show the utility of GRAFIMO by searching TFBS on a VG encoding the haplotypes from all the individuals sequenced by the 1000 Genomes Project (1000GP) [18,19].

## Design and implementation

GRAFIMO is a command-line tool, which enables a variant- and haplotype- aware search of TFBS, within a population of individuals encoded in a VG. GRAFIMO offers two main functionalities: the construction of custom VGs, from user data, and the search of one or more TF motifs, in precomputed VGs. Briefly, given a TF model (PWM) and a set of genomic regions, GRAFIMO leverages the VG to efficiently scan and report all the TFBS candidates and their frequency in the different haplotypes in a single pass together with the predicted changes in binding affinity mediated by genetic variations. GRAFIMO is written in Python3 and Cython and it has been designed to easily interface with the *vg* software suite [9]. For details on how to install and run GRAFIMO see **S1 Text** **section 7**.

## Genome variation graph construction

GRAFIMO provides a simple command-line interface to build custom genome variation graphs if necessary. Given a reference genome (FASTA format) and a set of genomic variants with respect to the reference (VCF format), GRAFIMO interfaces with the VG software suite to build the main VG data structure, the XG graph index [9] and the GBWT index [10,20] used to track the haplotypes within the VG. To minimize the footprint of these files and speedup the computation, GRAFIMO constructs the genome variation graph by building a VG for each chromosome. This also speeds-up the search operation since we can scan different chromosomes in parallel. Alternatively, the search can be performed one chromosome at the time for machines with limited RAM.

## Transcription factor motif search

The motif search operation takes as input a set of genomes encoded in a VG (.xg format), a database of known TF motifs (PWM in JASPAR [21] or MEME format [22]) and a set of genomic regions (BED format), and reports in output all the TFBS motifs occurrences in those regions and their estimated significance (**Fig 1**). To search for potential TFBS, GRAFIMO slides a window of length $k$ (where $k$ is the width of the query motif) along the paths of the VG corresponding to the genomic sequences encoded in it (**Fig 1B**). This is accomplished by an extension to the *vg find* function, which uses the GBWT index of the graph to explore the $k$-mer space of the graph while accounting for the haplotypes embedded in it [10]. By default, GRAFIMO considers only paths that correspond to observed haplotypes, however it is possible also to consider all possible recombinants even if they are not present in any individual. The significance (log-likelihood) of each potential binding site is calculated by considering the nucleotide preferences encoded in the PWM as in FIMO [13]. More precisely, the PWM is processed to a Position Specific Scoring Matrix (PSSM) (**Fig 1A**) and the resulting log-likelihood values are then scaled in the range [0, 1000] to efficiently calculate a statistical significance i.e. a $P$-value by dynamic programming [23] as in FIMO [13]. $P$-values are then converted to $q$-values by using the Benjamini-Hochberg procedure to account for multiple hypothesis testing. For this procedure, we consider all the $P$-values corresponding to all the $k$-mer-paths extracted within the scanned regions on the VG. GRAFIMO computes also the number of haplotypes in which a significant motif is observed and if it is present in the reference genome and/or in alternative genomes. (**Fig 1B**).

## Report generation

We have designed the interface of GRAFIMO based on FIMO, so it can be used as in-drop replacement for tools built on top of FIMO. As in FIMO, the results are available in three files:

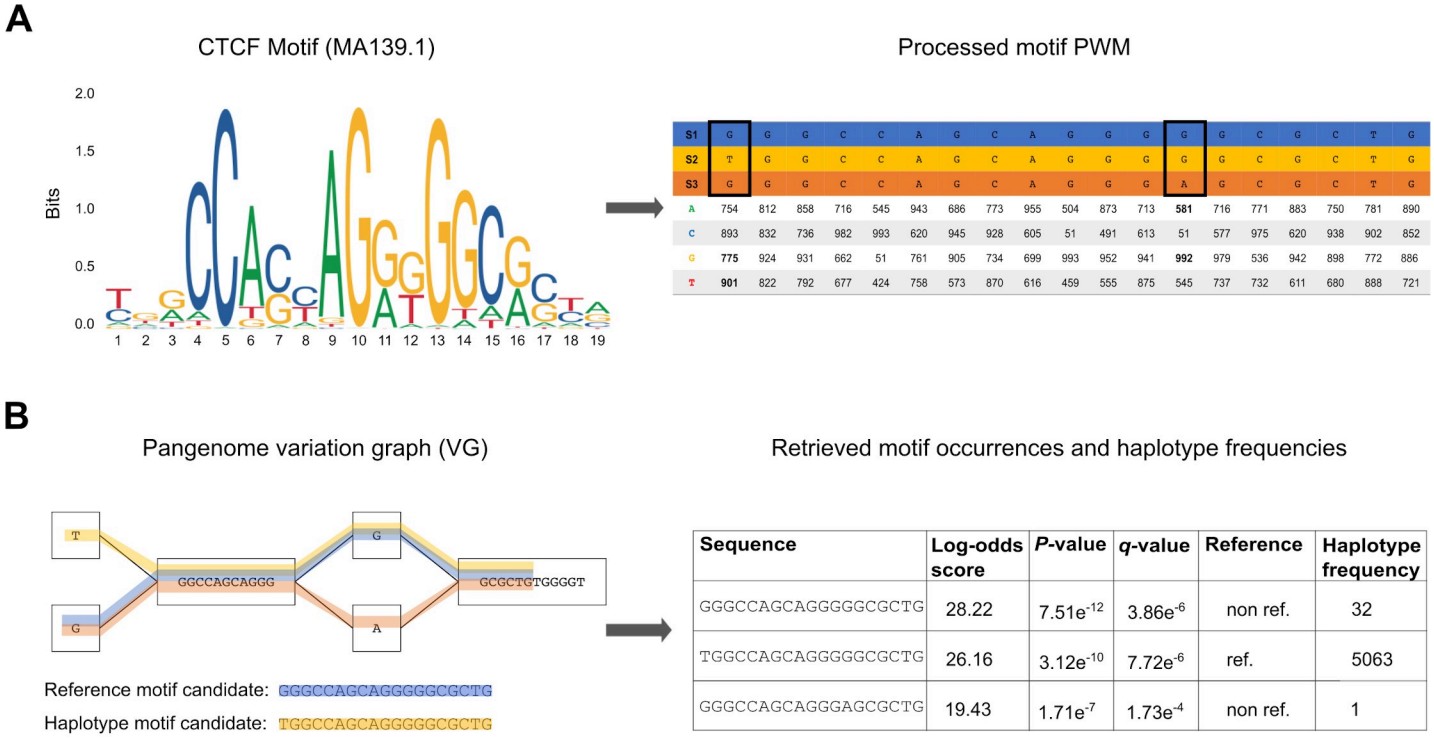

**Fig 1. GRAFIMO TF motif search workflow.** (A) The motif PWM (in MEME or JASPAR format) is processed and its values are scaled in the range [0, 1000]. The resulting score matrix is used to assign a score and a corresponding *P*-value to each motif occurrence candidate. In the final report GRAFIMO returns the corresponding log-odds scores, which are retrieved from the scaled values. (B) GRAFIMO slides a window of length *k*, where *k* is the motif width, along the haplotypes (paths in the graph) of the genomes used to build the VG. The resulting sequences are scored using the motif scoring matrix and are statistically tested assigning them the corresponding *P*-value and *q*-value. Moreover, for each entry is assigned a flag value stating if it belongs to the reference genome sequence ("ref") or contains genomic variants ("non.ref") and is computed the number of haplotypes in which the sequence appears.

a tab-delimited file (TSV), a HTML report and a GFF3 file compatible with the UCSC Genome Browser [24]. The TSV report (**Fig A** in **S1 Text**) contains for each candidate TFBS its score, genomic location (start, stop and strand), *P*-value, *q*-value, the number of haplotypes in which it is observed and a flag value to assess if it belongs to the reference or to the other genomes in VG. The HTML version of the TSV report (**Fig B** in **S1 Text**) can be viewed with any web browser. The GFF3 file (**Fig C** in **S1 Text**) can be loaded on the UCSC genome browser as a custom track, to visualize and explore the recovered TFBS with other annotations such as nearby genes, enhancers, promoters, or pathogenic variants from the ClinVar database [25].

## Results

As discussed above, GRAFIMO can be used to study how genetic variants may affect the binding affinity of potential TFBS within a set of individuals and may recover additional sites that are missed when considering only linear reference genomes without information about variants. To showcase its utility, we first constructed a VG based on 2548 individuals from the 1000GP phase 3 (hg38 human genome assembly) encoding their genomic variants and phased haplotypes (see **S1 Text** section 1 for details). We then searched this VG for putative TFBS for three TF motifs with different lengths (from 11 to 19 bp), evolutionary conservation, and information content from the JASPAR database [21]: CTCF (JASPAR ID MA0139.1), ATF3 (JASPAR ID MA0605.2) and GATA1 (JASPAR ID MA0035.4) (**Fig D** in **S1 Text**) (see **S1 Text**

**section 3-4-5**). To study regions with likely true binding events, for each factor we selected regions corresponding to peaks (top 3000 sorted by *q*-value) obtained by ChIP-seq experiments in 6 different cell types (A549, GM12878, H1, HepG2, K562, MCF-7) from the ENCODE project [26,27] (see **S1 Text** **section 2**). We used GRAFIMO to scan these regions and selected for our downstream analyses only sites with a *P*-value $< 1\mathrm{e}^{-4}$ and considered them as potential TFBS for these factors. Based on the recovered sites, we consistently observed across the 3 studied TFs that genetic variants can significantly affect estimated binding affinity. In fact, we found that thousands of CTCF motif occurrences are found only in non-reference haplotypes, suggesting that a considerable number of TFBS candidates are lost when scanning for TFBS the genome without accounting for genetic variants (**Fig 2A**). Similar results were

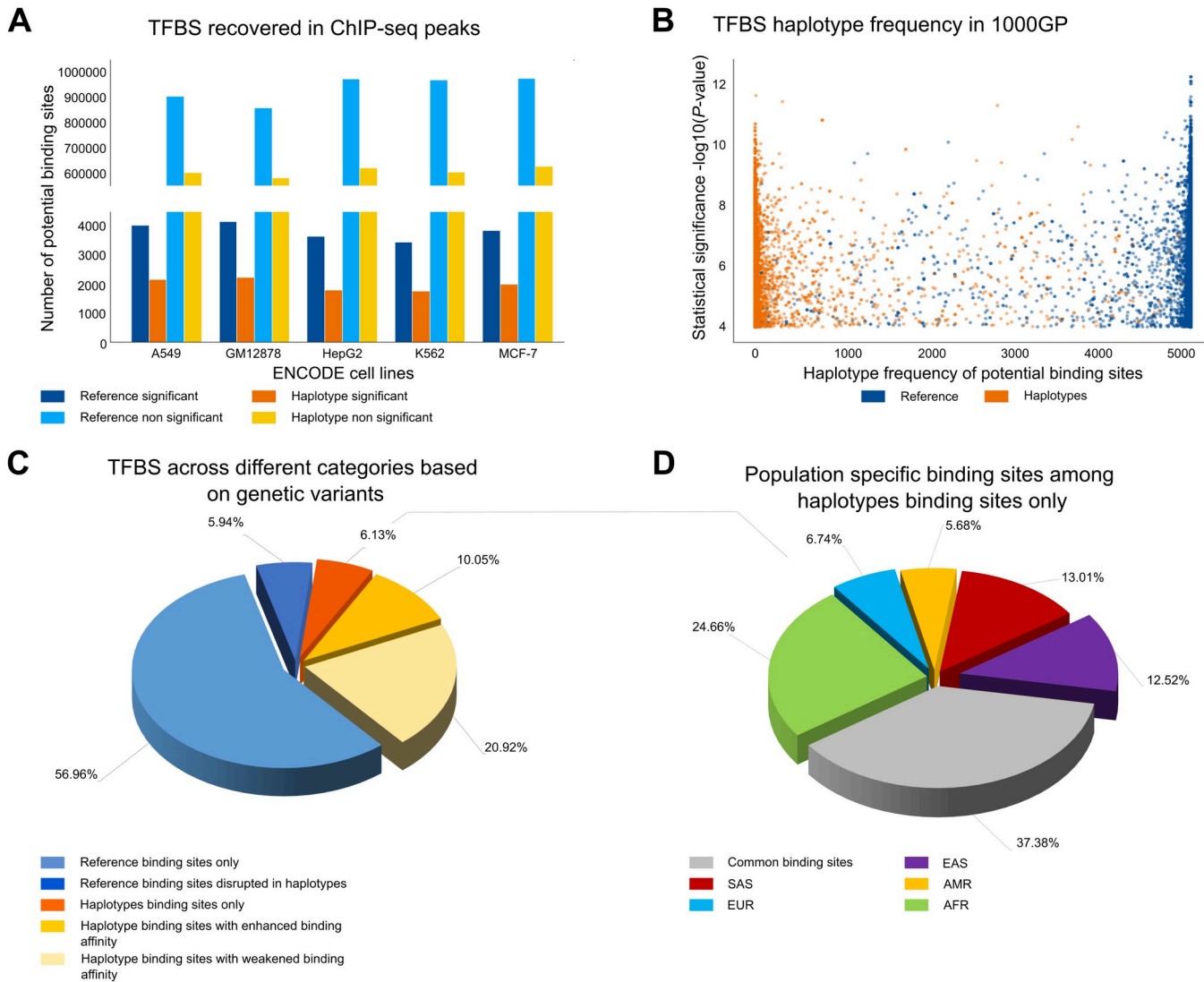

**Fig 2. Searching CTCF motif on VG with GRAFIMO provides an insight on how genetic variation affects putative binding sites.** (A) Potential CTCF occurrences statistically significant (P-value < 1e-4) and non-significant found in the reference and in the haplotype sequences found with GRAFIMO oh hg38 1000GP VG. (B) Statistical significance of retrieved potential CTCF motif occurrences and frequency of the corresponding haplotypes embedded in the VG. (C) Percentage of statistically significant CTCF potential binding sites found only in the reference genome or alternative haplotypes and with modulated binding scores based on 1000GP genetic variants (D) Percentage of population specific and common (shared by two or more populations) potential CTCF binding sites present on individual haplotypes.

obtained searching for ATF3 (**Fig E** in **S1 Text**) and GATA1 (**Fig F** in **S1 Text**). We also found several highly significant CTCF motif occurrences in rare haplotypes that may potentially modulate gene expression in these individuals (**Fig 2B**). Similar behaviours were observed for ATF3 (**Fig E** in **S1 Text**) and GATA1 (**Fig F** in **S1 Text**).

We also investigated the potential effects of the different length and type of mutations i.e. SNPs and indels on the CTCF, ATF3 and GATA1 binding sites. However, we did not observe a clear and general trend (**Fig G** in **S1 Text**). By considering the genomic locations of the significant motif occurrences we next investigated how often individual TFBS may be disrupted, created or modulated. We observed that 6.13% of the potential CTCF binding sites can be found only on non-reference haplotype sequences, 5.94% are disrupted by variants in non-reference haplotypes and ~30% are still significant in non-reference haplotypes but with different binding scores (**Fig 2C**). Similar results were observed for ATF3 (**Fig E** in **S1 Text**) and GATA1 (**Fig F** in **S1 Text**). Interestingly, we observed that a large fraction of putative binding sites recovered only on individual haplotypes are population specific. For CTCF we found that 24.66%, 6.74%, 5.68%, 13.01%, 12.52% of potential CTCF TFBS retrieved on individual haplotype sequences only are specific for AFR, EUR, AMR, SAS and EAS populations, respectively (**Fig 2D**). Similar results were observed for ATF3 and GATA1 (**S1 Text** **sections 4–5**).

Among the unique CTCF motif occurrences found only on non-reference haplotypes in CTCF ChIP-seq peaks we uncovered one TFBS (chr19:506,910–506,929) that clearly illustrates the danger of only using reference genomes for motif scanning. Within this region we recovered a heterozygous SNP that overlaps (position 10 of the CTCF matrix) and significantly modulates the binding affinity of this TFBS. In fact, by inspecting the ChIP-seq reads (experiment ENCSR000DZN, GM12878 cell line), we observed a clear allelic imbalance towards the alternative allele G (70.59% of reads) with respect to the reference allele A (29.41% of reads). This allelic imbalance is not observed in the reads used as control (experiment code ENCSR000EYX) (**Fig 3**).

Taken together these results highlight the importance of considering non-reference genomes when searching for potential TFBS or to characterize their potential activity in a population of individuals.

We also compared the performance of GRAFIMO against FIMO [13] (**Fig H** in **S1 Text** and **S1 Text** **Section 6**). FIMO is faster and requires less memory, when scanning a single linear genome. However, when considering the 2548 individual genomes and their genetic variation, GRAFIMO proves to be generally faster than FIMO. Moreover, we benchmarked how GRAFIMO running time and memory usage change using an increasing number of threads (**Fig I** in **S1 Text**). By increasing the number of threads, we observed a dramatical drop in running time, while memory usage remained similar.

## Conclusion

By leveraging VGs, GRAFIMO provides an efficient method to study how genetic variation affects the binding landscape of a TF within a population of individuals. Moreover, we show that several potential and private TFBS are found in individual haplotype sequences and that genomic variants significantly also affect the binding affinity of several motif occurrence candidates found in the reference genome sequence. Our tool therefore can help in prioritizing potential regions that may mediate individual specific changes in gene expression, which may be missed by using only reference genomes.

## Availability and future directions

GRAFIMO can be downloaded and installed via PyPI, source code or Bioconda. Its Python3 source code is available on Github at https://github.com/pinellolab/GRAFIMO and at https://

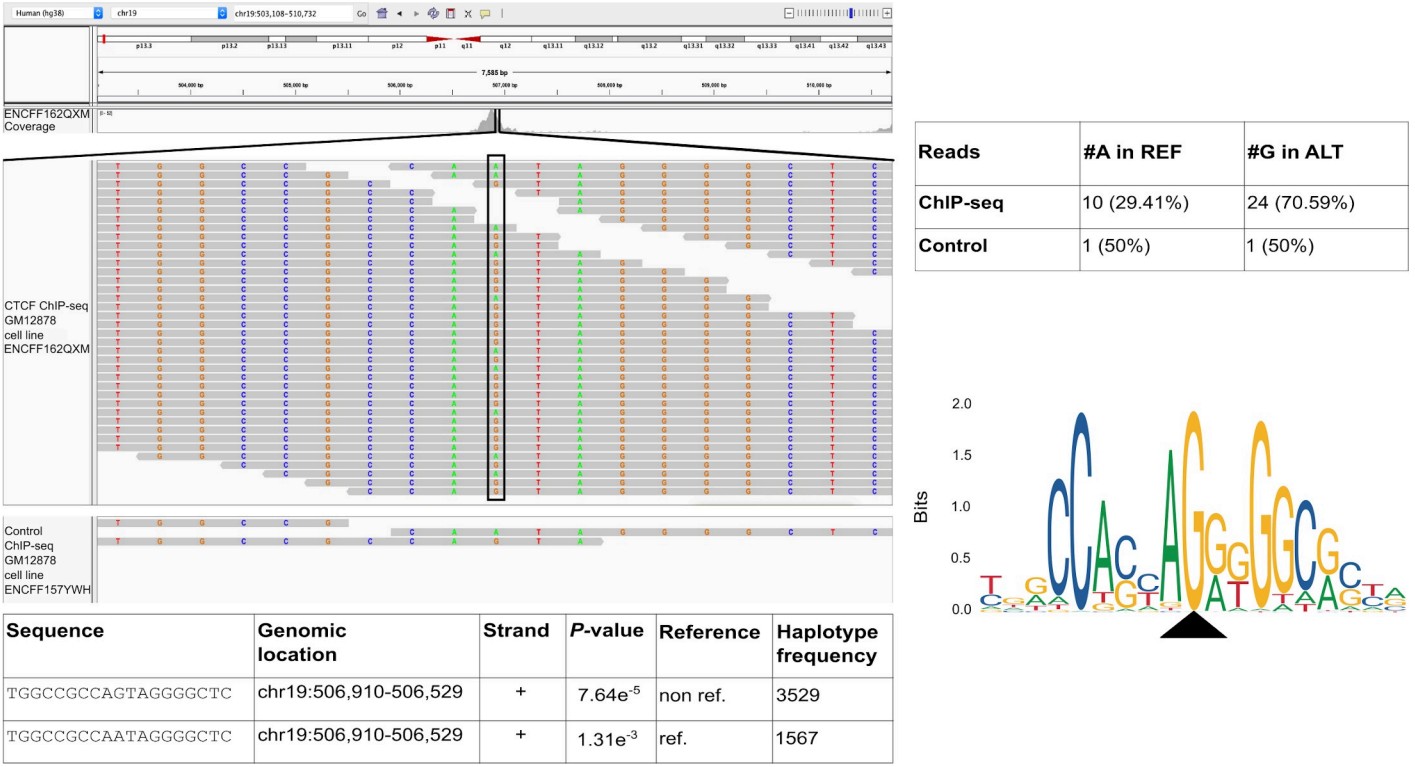

| Reads | #A in REF | #G in ALT |
|---|---|---|
| ChIP-seq | 10 (29.41%) | 24 (70.59%) |
| Control | 1 (50%) | 1 (50%) |

| Sequence | Genomic location | Strand | P-value | Reference | Haplotype frequency |
|---|---|---|---|---|---|
| TGGCCGCCAGTAGGGGCTC | chr19:506,910-506,529 | + | $7.64e^{-5}$ | non ref. | 3529 |
| TGGCCGCCAATAGGGGCTC | chr19:506,910-506,529 | + | $1.31e^{-3}$ | ref. | 1567 |

**Fig 3. Considering genomic variation, GRAFIMO captures more potential binding events.** GRAFIMO reports a potential CTCF binding site at chr19:506,910–506,929 found only in haplotype sequences, searching the motif in ChIP-seq peaks called on cell line GM12878 (experiment code ENCSR000DZN). The reads used to call for ChIP-seq peaks (ENCFF162QXM) show an allelic imbalance at position 10 of the motif sequence towards the alternative allele G, instead of the reference allele A. The imbalance is captured by GRAFIMO which reports the sequence presenting G at position 10 (found in the haplotypes), while the potential TFBS on the reference carrying an A is not reported as statistically significant (P-value $> 1e^{-4}$). CTCF motif logo shows that the G is the dominant nucleotide in position 10.

github.com/InfOmics/GRAFIMO under MIT license. Since GRAFIMO is based on VG data structure, has the potential to be applied to future pangenomic reference systems that are currently under development (https://news.ucsc.edu/2019/09/pangenome-project.html). The genome variation graphs enriched with 1000GP on GRCh38 phase 3 used to obtain the results presented in this manuscript can be downloaded at http://ncrnadb.scienze.univr.it/vgs.

## Supporting information

**S1 Text. Additional information about experiments design, GATA1 and ATF3 search on genome variation graph, and how to install and run GRAFIMO. Fig A. Example of TSV summary report.** The tab-delimited report (TSV report) shows the first 25 potential CTCF occurrences retrieved by GRAFIMO, searching the motif in ChIP-seq peak regions defined in ENCODE experiment ENCFF816XLT (cell line A549). **Fig B. Example of HTML summary report.** The HTML report shows the first 25 potential CTCF occurrences retrieved by GRA-FIMO, searching the motif in ChIP-seq peak regions defined in ENCODE experiment ENCFF816XLY (cell line A549). **Fig C. Example of GFF3 track produced by GRAFIMO, loaded on the UCSC genome browser.** GRAFIMO returns also a GFF3 report which can be loaded on the UCSC genome browser; the loaded custom track shows three potential CTCF occurrences (region chr8:142,782,661–142,782,680) retrieved by GRAFIMO overlapping a dbSNP annotated variant (rs892844) (image obtained from the UCSC Genome Browser website). **Fig D. Structure of transcription factor motifs used to test GRAFIMO.** Transcription

factor binding site motifs of (A) CTCF, (B) ATF3 and (C) GATA1. **Fig E. Searching ATF3 motif on VG with GRAFIMO provides an insight on how genetic variation affects the binding site sequence.** (A) Potential ATF3 occurrences statistically significant (P-value $< 1e^{-4}$) and non-significant found in the reference and in the haplotype sequences found with GRAFIMO oh hg38 1000GP VG. (B) Statistical significance of retrieved potential ATF3 motif occurrences and their frequency in the haplotypes embedded in the VG. (C) Percentage of statistically significant ATF3 potential binding sites found only in genome reference sequence, percentage of potential TFBS found in the reference for which genetic variants cause the sequence to be no more significant, percentage of binding sites found only in the haplotypes, percentage of potential TFBS found in the reference with increased statistical significance by the action of genomic variants and percentage of those with a decreased significance by the action of variants (with P-value still significant). (D) Fraction of population specific potential ATF3 binding sites recovered on individual haplotype sequences. **Fig F. Searching GATA1 motif on VG with GRAFIMO provides an insight on how genetic variation affects the binding site sequence.** (A) Potential GATA1 occurrences statistically significant (P-value $< 1e^{-4}$) and non-significant found in the reference and in the haplotype sequences found with GRAFIMO oh hg38 1000GP VG. (B) Statistical significance of retrieved potential GATA1 motif occurrences and their frequency in the haplotypes embedded in the VG. (C) Percentage of statistically significant GATA1 potential binding sites found only in genome reference sequence, percentage of potential TFBS found in the reference for which genetic variants cause the sequence to be no more significant, percentage of binding sites found only in the haplotypes, percentage of potential TFBS found in the reference with increased statistical significance by the action of genomic variants and percentage of those with a decreased significance by the action of variants (with *P*-value still significant). (D) Percentage of population specific potential GATA1 binding sites, among those TFBS retrieved uniquely on individual genome sequences. **Fig G. Influence of the different length and type of mutations on binding affinity score**. (A) CTCF, (B) ATF3, (C) GATA1. **Fig H. Comparing GRAFIMO and FIMO performance.** (A) Searching CTCF motif (JASPAR ID MA0139.1) on human chr22 regions (total width ranging from 1 to 9 millions of bp) without accounting for genetic variants FIMO is faster than GRAFIMO (using a single thread). (B) FIMO uses less memory resources than GRAFIMO, however they work on different frameworks. (C) When considering the genetic variation present in large panels of individuals as 1000GP on GRCh38 phase 3 (2548 samples), GRAFIMO proves to be faster than FIMO in searching potential CTCF occurrences. It is faster when run with a single execution thread, and significantly faster when run with 16. **Fig I. GRAFIMO running time efficiently scales with the number of threads used.** By running GRAFIMO with multiple threads (A) the running time significantly decreases, while (B) memory usage remains similar. **Table A. Number of genomic variants used to test GRAFIMO.** Number of genomic variants used to test GRAFIMO, divided by chromosome. The variants were obtained from 1000 Genomes Project on GRCh38 phase 3, and belongs to 2548 individuals from 26 populations. The number of variants refers to SNPs and indels together. In total were considered ~78 million variants. **Table B. ENCODE ChIP-seq experiment codes.** To test our software, we searched the potential occurrences of three transcription factor motifs (CTCF, ATF3 and GATA1) in a hg38 pangenome variation graph enriched with genomic variants and haplotypes of 2548 individuals from 1000 Genomes project phase 3. To have likely to happen binding events, TF motifs were searched in ChIP-seq peak regions, obtained from the ENCODE project data portal.
(DOCX)

**S1 Code. GRAFIMO v1.1.4 source code, documentation and running examples.**
(ZIP)

## Acknowledgments

We would like to thank Centro Piattaforme Tecnologiche (CPT) located in the University of Verona that provided us with all the hardware necessary to perform all the tests. Research reported in this publication was supported by the National Human Genome Research Institute of the National Institutes of Health under Award Number R00HG008399 and Genomic Innovator Award Number R35HG010717. The content is solely the responsibility of the authors and does not necessarily represent the official views of the National Institutes of Health.

## Author Contributions

**Conceptualization:** Rosalba Giugno, Luca Pinello.

**Formal analysis:** Manuel Tognon.

**Methodology:** Manuel Tognon, Vincenzo Bonnici, Erik Garrison, Rosalba Giugno, Luca Pinello.

**Software:** Manuel Tognon, Erik Garrison.

**Supervision:** Rosalba Giugno, Luca Pinello.

**Validation:** Luca Pinello.

**Writing – original draft:** Manuel Tognon.

**Writing – review & editing:** Manuel Tognon, Vincenzo Bonnici, Erik Garrison, Rosalba Giugno, Luca Pinello.

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
