## [Decision Letter · Decision Letter 0]

7 Mar 2021

Dear Dr. Pinello,

Thank you very much for submitting your manuscript "GRAFIMO: variant and haplotype aware motif scanning on pangenome graphs" for consideration at PLOS Computational Biology.

As with all papers reviewed by the journal, your manuscript was reviewed by members of the editorial board and by several independent reviewers. In light of the reviews (below this email), we would like to invite the resubmission of a significantly-revised version that takes into account the reviewers' comments.

We cannot make any decision about publication until we have seen the revised manuscript and your response to the reviewers' comments. Your revised manuscript is also likely to be sent to reviewers for further evaluation.

Sincerely,

Mihaela Pertea

Software Editor

PLOS Computational Biology

Mihaela Pertea

Software Editor

PLOS Computational Biology

Reviewer's Responses to Questions

**Comments to the Authors:**

Reviewer #1: The authors presented a new tool to scan known TF DNA motifs in VG (graph genome). The manuscript overall is clear and the description is concise. The idea is not fully novel, for example similar study is reported here: (Cristian Groza, et. al. 2020 Genome Biology). Also, the tool itself heavily based on VG and overall is simple, but it do have many potential users.

I have the following comments to help improve the manuscript:

1. The authors construct the VG from the 1000G data, which is a diverged dataset. Are there population specific TF motifs are found? What’s the ratio of that?

2. For those discovered non-ref TF motifs, what’s the effect from different type/length of mutations? For example, do indel and larger mutations have larger effect?

3. The authors need to benchmark the performance (memory/running time) of the tool.

Reviewer #2: This is a well written manuscript, presenting a method that I think will be useful for many researchers in the field. GRAFIMO is a nice application of genome graphs, and the method provides a clear benefit compared to the existing "linear" method FIMO. I particularly like that you've designed GRAFIMO so that it can be used as a direct replacement to FIMO. I don't have many comments and I really don't have any major issues with the manuscript. However, I ran into some problems when trying to run the software, and would like you to fix these so that I'm able to fully try out the software before concluding on the manuscript. These are my comments:

- I have a few questions related to how you do false discovery rate adjustment of the p-values for each motif match. As far as I understand Fimo, q-values are found simply by accounting for the number of hypothesis tests done within the specified regions on the linear reference genome (i.e. one test for every base pair within the regions). I don't find any information in the manuscript or in the supplementary text on how you do this on a graph. I assume that you maybe account for all the hypothesis tests performed on the graph, i. e. the number of kmer-paths in the graph (following known haplotypes, if specified)? I know that FIMO has to hold all matches in memory in order to compute these p-values, and that the user can specify how many to hold in memory through the --max-stored-scores option. I don't see this option in GRAFIMO.

- Installation of Grafimo went fine on my system (used pip). I only had a minor issue with sphinx not being specified as a dependency in the setup.py file. Should that be added?

- How does GRAFIMO compare to FIMO when it comes to run time? Is it considerably slower than FIMO, since it has to search for more potential motif matches? I personally don't mind if it is a bit slower than FIMO, but it would be nice if you could include a sentence about runtime. If runtime isn't an issue, that could be mentioned, and if runtime is an issue, it could be nice for the reader to know for instance how long time GRAFIMO and FIMO spends on processing e.g. one chromosome using one thread.

- If I only run "grafimo" on the command line, I get an IndexError (probably because I didn't specify -h or --help or anything else). It would be nice to instead get a more user friendly error, or just the help message.

- I tried running GRAFIMO on some of my own data, and I think I ran into some issues when running "grafimo buildvg" since it assumed I had chromosome chr1, chr2, chr3 and so on. My test dataset only had one chromosome "1" (not "chr1"). So I tried running "grafimo buildvg -h" to see if I could specify the chromosomes for my data. However, it seems that this doesn't give me the options for the subcommand "buildvg", but all the options for GRAFIMO? Or am I wrong? Anyway, it was a bit unclear for me how I could see which arguments only the buildvg command has.

- After specifying -c 1 to grafimo buildvg to try to make it only build graphs for chromosome 1, it seems that it tries to build a graph for "chr1". Would it maybe be better to let the user be able to specify the exact literal chromosomes, so that the user would need to specify "chr1" if chr1 is wanted (at least I often have data without the chr prefix). Right now it seems that chromosome "1" and so one (without chr) is not supported? EDIT: After I got another error (see next point), I tried installing the latest GRAFIMO through github, and it seems that things have changed there (from what is in the pip package)? On line 507 in constructVG.py, it seems that you remove "chr" from chromosome names, so when I now use "chr1" in my data, GRAFIMO crashes with "ValueError: Unknown chromosome given". Unless I have misunderstood something, it would be good if you fixed these issues. As a user, I would ideally be able to specify either "chr1" or "1", and GRAFIMO should then support both (and not convert the chromosome names).

- After solving the issues with chromosome by using other data (now data with "chr1"), I got the following error message when using the example.meme file:

File "/usr/local/lib/python3.8/dist-packages/grafimo/motif.py", line 634, in read_MEME_motif

motifID, motifName = line.split()[1:3]

ValueError: not enough values to unpack (expected 2, got 1)

When checking line 634 in motif.py in your github-repo, I realised that this might be an old bug that is fixed since the pip-package was published. That's when I decided to pull your latest code from Github, and I then ran into issues with "chr1" not being supported anymore. At this point I gave up, but I guess there's not much that needs to be fixed on your side before I should be able to run GRAFIMO. Let me know if you believe I've misunderstood anything or done anything wrong.

- I think it is very nice that you have included so many details about the experiments in the supplementary text, but I don't find any scripts for reproducing the experiments. It would be nice with reference to scripts (and ideally also the graphs) you used for running the experiments.

**Have all data underlying the figures and results presented in the manuscript been provided?**

Reviewer #1: Yes

Reviewer #2: Yes

PLOS authors have the option to publish the peer review history of their article (what does this mean?). If published, this will include your full peer review and any attached files.

Reviewer #1: No

Reviewer #2: **Yes: **Ivar Grytten
---

## [Decision Letter · Decision Letter 1]

14 Jun 2021

Dear Dr. Pinello,

Thank you very much for submitting your manuscript "GRAFIMO: variant and haplotype aware motif scanning on pangenome graphs" for consideration at PLOS Computational Biology. As with all papers reviewed by the journal, your manuscript was reviewed by members of the editorial board and by several independent reviewers. The reviewers appreciated the attention to an important topic. Based on the reviews, we are likely to accept this manuscript for publication, providing that you modify the manuscript according to the review recommendations.

Sincerely,

Mihaela Pertea

Software Editor

PLOS Computational Biology

Mihaela Pertea

Software Editor

PLOS Computational Biology

[LINK]

Reviewer's Responses to Questions

**Comments to the Authors:**

Reviewer #1: I finished reviewing the revised manuscript. The authors have answered all my previously proposed questions, and the work is ready to be accepted. It’s a nice tool and will potentially have many users!

Reviewer #2: - I think the experiments you have performed for examining running time and memory usage are satisfactory. In the end of the section titled "Transcription factor motif search" you have added a sentence referring to these experiments. I find it a bit weird to refer to the experiments here. Wouldn't this be better suited for the Results section? The way you phrase this sentence is also a bit vague. You say "S1 Text section 6 describes how the tool performance compares to those of a tool designed to work on a single genomic sequence at a time, such as FIMO". I think two things could be improved here: 1) You could just say that you compare the performance to Fimo, instead of "tools, such as FIMO" (since you only compare to FIMO) and 2) it would be nice if you could very briefly summarize the main findings here, instead of just referring to the supplementary. What I would prefer is a phrasing along the lines of "We also compared the performance of GRAFIMO against FIMO, and found that ... (see S1 Text Section 6).".

- In the caption of S9 Figure you say that memory usage increases when using multiple threads. But from the figure, it seems that the memory usage is about the same for all the cases. Am I right or have I misunderstood the figure?

- You say that you have made all scripts for reproducing the experiments available in a dedicated Github repository, but in your comment you link to the GRAFIMO Github repository. Is this the wrong link? I cannot seem to find the shell scripts for reproducing the experiments in this repository.

**Have the authors made all data and (if applicable) computational code underlying the findings in their manuscript fully available?**

Reviewer #1: Yes

Reviewer #2: **No: **

PLOS authors have the option to publish the peer review history of their article (what does this mean?). If published, this will include your full peer review and any attached files.

Reviewer #1: **Yes: **Chong Chu

Reviewer #2: **Yes: **Ivar Grytten

Figure Files:

Data Requirements:

Reproducibility:

References:

---

## [Decision Letter · Decision Letter 2]

10 Sep 2021

Dear Dr. Pinello,

We are pleased to inform you that your manuscript 'GRAFIMO: variant and haplotype aware motif scanning on pangenome graphs' has been provisionally accepted for publication in PLOS Computational Biology.

Best regards,

Mihaela Pertea

Software Editor

PLOS Computational Biology

Feilim Mac Gabhann

Editor-in-Chief

PLOS Computational Biology

Reviewer's Responses to Questions

**Comments to the Authors:**

Reviewer #2: The atuhors have satisfactorily addressed all my comments, and I am now very happy with the manuscript and accept it for publication.

**Have the authors made all data and (if applicable) computational code underlying the findings in their manuscript fully available?**

Reviewer #2: Yes

PLOS authors have the option to publish the peer review history of their article (what does this mean?). If published, this will include your full peer review and any attached files.

Reviewer #2: **Yes: **Ivar Grytten

---

## [Editor Report · Acceptance letter]

22 Sep 2021

PCOMPBIOL-D-21-00041R2 

GRAFIMO: variant and haplotype aware motif scanning on pangenome graphs

Dear Dr Pinello,

I am pleased to inform you that your manuscript has been formally accepted for publication in PLOS Computational Biology. Your manuscript is now with our production department and you will be notified of the publication date in due course.

With kind regards,

Zsofi Zombor
